

# New particle formation events observed at King Sejong Station, Antarctic Peninsula – Part 1: Physical characteristics and contribution to cloud condensation nuclei

Jaeseok Kim[1,2], Young Jun Yoon[1,*], Yeontae Gim[1], Jin Hee Choi[1], Hyo Jin Kang[1,3], Ki-Tae Park[1], Jiyeon Park[1], and Bang Yong Lee[1]

[1]Korea Polar Research Institute, 26 Songdomirae-ro, Yeonsu-gu, Incheon 21990, Republic of Korea

[2]Korea Research Institute of Standards and Science, 267 Gajeong-ro, Yuseong-gu, Daejeon 34113, Republic of Korea

[3]University of Science & Technology (UST), 217 Gajeong-ro, Yuseong-gu, Daejeon 34113, Republic of Korea

[*]*Correspondence to*: Young Jun Yoon (yjyoon@kopri.re.kr)

**Abstract**

The physical characteristics of aerosol particles during a particle burst observed at King Sejong Station in Antarctic Peninsula from March 2009 to December 2016 were analyzed. This study focuses on the seasonal variation in parameters related to particle formation such as the occurrence, formation rate (FR) and growth rate (GR), condensation sink (CS), and source rate of condensable vapor. The number concentrations during new particle formation (NPF) events varied from 1707 cm$^{-3}$ to 83120 cm$^{-3}$, with an average of 20649 ± 9290 cm$^{-3}$, and the duration of the NPF events ranged from 0.6 h to 14.4 h, with a mean of 4.6 ± 1.5 h. The NPF event dominantly occurred during austral summer period (~72%). The mean values of FR and GR of the aerosol particles were 2.79 ± 1.05 cm$^{-3}$ s$^{-1}$ and 0.68 ± 0.27 nm h$^{-1}$, respectively showing enhanced rates in the summer season. The mean value of FR at King Sejong Station was higher than that at other sites in Antarctica, at 0.002-0.3 cm$^{-3}$ s$^{-1}$, while those of growth rates was relatively similar results observed by precious studies, at 0.4~4.3 nm h$^{-1}$. The average values of CS and source rate of condensable vapor were (6.04 ± 2.74) × 10$^{-3}$ s$^{-1}$ and (5.19 ± 3.51) × 10$^{4}$ cm$^{-3}$ s$^{-1}$, respectively. The contribution of particle formation to cloud condensation nuclei (CCN) concentration was also investigated. The CCN concentration during the NPF period increased





approximately 9% compared with the background concentration. In addition, the effects of the origin
and pathway of air masses on the characteristics of aerosol particles during a NPF event were
determined. The FRs were similar regardless of the origin and pathway, whereas the GRs of particles
originating from the Antarctic Peninsula and the Bellingshausen Sea, at $0.77 \pm 0.25$ nm h$^{-1}$ and $0.76 \pm$
$0.30$ nm h$^{-1}$, respectively, were higher than those of particles originating from the Weddell Sea ($0.41 \pm$
$0.15$ nm h$^{-1}$).
**1. Introduction**
Understanding the effect of atmospheric aerosol particles on climate change is an important issue
in atmospheric science. These particles are highly significant substances in the radiation transfer
process in the atmosphere, with direct effects through scattering and absorption of solar radiation and
indirect effects by acting as cloud condensation nuclei (CCN) for cloud droplets (Anttila et al., 2012).
These particles also influence the properties and life time of clouds (Twomey, 1977; Albrecht, 1989).
Although aerosol particles play an important role in global and regional climates, large uncertainties
remain owing to a lack of knowledge on their formation and physicochemical characteristics (Carslaw
et al., 2013; IPCC, 2013).
New particle formation (NPF) frequently occurs in the atmosphere and leads to enhancement of the
total number concentrations of aerosol particles due to high numbers of nucleation mode particles
(Spracklen et al., 2006; Dallósto et al., 2017). The modeling study of Pierce and Adams (2007)
indicates that new ultrafine particles of <100 nm can contribute to maximum CCN generations of 40%
and 90% at the boundary layer and in the remote free troposphere, respectively. In order to understand
the characteristics of the NPF, studies have been conducted in various regions including coastal, forest,
mountainous, rural and urban sites (O'Dowd et al., 2002; Komppula et al., 2003; Kulmala et al., 2004;
Yoon et al., 2006; Park et al., 2009; Kim et al., 2011; Rose et al., 2015; Bianchi et al., 2016; Kontkanen
et al., 2017). In addition, studies on the NPF phenomenon have recently been conducted at various





sites in the polar regions (Asmi et al., 2010; Järvinen et al., 2013; Kyrö et al., 2013; Park et al., 2004;
Weller et al., 2015; Humphries et al., 2016; Nguyen et al., 2016; Willis et al., 2016; Barbaro et al.,
2017; Dallósto et al., 2017). A NPF event occurring form December 1998 to December 2000 at the
South Pole was reported by Park et al. (2004). Kyrö et al. (2013) showed that oxidized organics derived
from the oxidation of biogenic precursors originating from local melting ponds might have contributed
to particle growth at the Finnish research station Aboa (73.50°S, 13.42°W). In addition, studies on the
NPF were conducted at the Concordia station, Dome C (75.10°S, 123.38°E; Järvinen et al., 2013) and
at the coastal Antarctic station Neumayer (70.65°S, 8.25°W; Weller et al., 2015). Although studies on
NPF events have been conducted at various stations in the Antarctica, no results are available for the
station in the Antarctic Peninsula. Also, the contribution of NPF to CCN concentration is not well
understood in this area. Furthermore, results of the general long-term characteristics of aerosol
particles during the period of NPF observation in Antarctica are rare compared with those in other
continents.
In the present study, the frequency of NPF events was determined on the basis of total aerosol
number concentration. We investigated the physical characteristics such as formation rate (FR) and
growth rate (GR), condensation sink (CS) and source of condensation vapor as well as the seasonality
of atmospheric aerosols during NPF events at King Sejong Station in the Antarctic Peninsula. The
effect of particle formation on CCN concentrations was also examined. Furthermore, the air mass back
trajectories were analyzed by using the Hybrid Single Particle Lagrangian Integrated Trajectory
(HYSPLIT) model to understand physical properties of NPF events depending on the origins and
pathway of the air masses.
**2. Methods**
**2.1. Site description and instrumentation**
The data analyzed in this study were obtained at the King Sejong station in the Antarctic Peninsula



(62.22°S, 58.78°W). Further details on the sampling site as well as the instrumental set-up were
introduced in our previous study (Kim et al., 2017). In brief, two condensation particle counters (CPCs;
TSI 3776 and TSI 3772) were used to measure the total particle number concentrations. The size
distributions of the aerosol particles ranging from 10 nm to 300 nm were measured by using a scanning
mobility particle sizer (SMPS), which combined a differential mobility analyzer (DMA; HCT Inc.,
LDMA 4210) and a CPC (TSI 3772). The CCN concentrations were determined by using a CCN
counter (CCNC; DMT CCN-100). In addition, meteorological parameters including temperature,
relative humidity, wind speed, wind direction, pressure, and solar radiation intensity were continuously
monitored by using an automatic weather station (AWS; Vaisala HMP45 for measuring temperature
and relative humidity, WeatherTronics 2102 for measuring wind speed and direction, WeatherTronics
7100 for measuring pressure and Eppley Precision Spectral Pyranometer PSP for measuring solar
radiation intensity) system.
**2.2. Data analysis**
To ensure data quality, raw data of the following conditions were discarded: (i) wind direction
between 355° and 55° (local pollution sector) (ii) concentration of black carbon higher than 100 ng m$^{-}$
$^{3}$, (iii) wind speed less than 2 m s$^{-1}$ and (iv) instrument malfunction based on the log-book. If valid data
for one day were less than 50% after discarding the raw data, such days were excluded. The acquisition
rate for each instrument is summarized in Table 1. Here, the acquisition rate indicates the value of the
analyzed days divided by the total measurement days. Because the acquisition rate from the SMPS
was lower than that of the CPC in this study, the value difference between the concentrations of
particles larger than 2.5 nm ($CN_{2.5}$) and 10 nm ($CN_{10}$) observed from two CPCs was used to identify
the NPF events.
**2.2.1. Definition of NPF events**
As mentioned in the previous section, the value difference between $CN_{2.5}$ and $CN_{10}$ concentrations





were used to define days for NPF events or non-NPF events (Yoon et al., 2006). The value difference
($CN_{2.5}$-$CN_{10}$) represents the number concentrations of newly formed particles produced from gas-to-
particle conversion. The NPF days were defined in this study according to the following conditions:
(i) The difference in number concentrations ($CN_{2.5}$-$CN_{10}$) is higher than 500 $cm^{-3}$ (ii) the ($CN_{2.5}$-
$CN_{10}$)/$CN_{10}$ ratio is higher than 10 and (iii) the NPF duration is longer than 30 min. The ($CN_{2.5}$-
$CN_{10}$)/$CN_{10}$ ratio is the parameter used to distinguish between particles newly formed from gas-to-
particle conversion and background particles (Warren and Seinfeld, 1985; Humphries et al., 2015).
Humphries et al. (2016) also used the ($CN_{2.5}$-$CN_{10}$)/$CN_{10}$ ratio to distinguish the NPF days during a 52
days voyage in the East Antarctic sea ice region because the number concentration data were more
reliable than the size distribution data.
**2.2.2. Classification of NPF events using SMPS data**
After identification of the NPF event days, classification of the NPF events was conducted by using
size distributions from a SMPS. The NPF events were classified into three types of A, B and C
according to the classification by Dal Maso et al. (2005) and Yli-Juuti et al. (2009). Type A describes
days in which the formation and growth of particles were clear. Type B describes days in which the
formation occurred but growth was not clear. Type C describes days in which the event occurrence
was not distinct.
**2.2.3. Estimation of parameters for NPF characteristics**
On the basis of the number concentration data with 1 s time resolution the FR was calculated for
cases in which the concentrations of ($CN_{2.5}$-$CN_{10}$) sharply increased. The FR of new particles ranging
from 2.5 nm to 10 nm was determined according to variation in the number concentrations of $CN_{2.5-10}$
($CN_{2.5-10}$=$CN_{2.5}$-$CN_{10}$) based on the following equation (Dal Maso et al., 2005):

$$FR = \frac{dN_{nuc}}{dt} + F_{coag} + F_{growth} \qquad (1)$$



Here, $N_{nuc}$ is the particle number concentrations of nucleation mode. In this study, the $CN_{2.5-10}$
concentrations obtained by two particle counters were used for the term $N_{nuc}$. $F_{coag}$ is the particle loss
in accordance with coagulation, and $F_{growth}$ represents the flux of particles growing from the nucleation
mode. Because the $CN_{2.5-10}$ concentrations were predominant in the total number concentration and
the particles rarely grew over the nucleation mode during the formation period, the $F_{coag}$ and $F_{growth}$
terms in Eq. 1 were neglected in this study (Dal Maso et al., 2005; Shen et al., 2016).
The GRs were calculated by using the size distributions measured by a SMPS. In order to calculate
the geometric mean diameter (GMD) as a function of time, we chose the particle range of 10-20 nm
due to the size resolution of the SMPS. The GR was determined by rate of change in the GMD by
using the following equation (Kulmala et al., 2004; Dal Maso et al., 2005):

$$GR = \frac{dD_p}{dt} \qquad (2)$$

The CS is an important parameter governing the NPF because it indicates the speed in which
gaseous molecules condense onto pre-existing aerosols. It can be estimated from the size distribution
data according to the following equation (Dal Maso et al., 2005; Kulmala et al., 2005; Shen et al.,

18 2016):

$$CS = 2\pi D \sum_{dp} \beta_m d_p N_{dp} \qquad (3)$$

where $D$ is the diffusion coefficient of the condensable vapor, $\beta$ is the transitional regime correction
factor from Fuchs and Sutugin (1970), and $d_p$ and $N_{dp}$ are the particle size and number concentration,
respectively. It is assumed that condensable vapor is gaseous sulfuric acid which has been reported to
play an important role in the nucleation process (Dal Maso et al., 2005).





According to the GR and the CS, it is possible to estimate condensable vapor concentration, $C_v$ (unit:
molecules $cm^{-3}$) and its source rate, $Q$ (unit: molecules $cm^{-3}$ $s^{-1}$; Kulmala et al., 2001; Dal Maso, 2002),
assuming that the particle growth is caused by condensation of a low volatile vapor to the particle
surface. In the nucleation mode, the relationship between $C_v$ and GR is estimated by the following
equation:
$$C_v = A \times GR \qquad (4)$$
where $A$ is a constant, specifically $1.37 \times 10^7$ h $cm^{-3}$ for a vapor with the molecular properties of sulfuric
acid. It assumed that $C_v$ is constant during the growth process.
Assuming no other sink terms for the condensing vapor, source rate of condensable vapor is
estimated under the steady-state condition:
$$Q = CS \times C_v \qquad (5)$$
**2.3. Backward trajectory analysis**
To understand characteristics of NPF events depending on the origin and pathway of air masses, air
mass backward trajectory analysis was performed by using the HYSPLIT model (Stein et al., 2015;
http://www.arl.noaa.gov/HYSPLIR.php). Typical 48-h air mass backward trajectories were analyzed,
ending at heights of 100m, 500m, and 1500m above the ground level of the sampling site. The results
with similar air mass origins and pathways during the NPF periods at three different heights were used
for the analysis in this study. Accordingly, the air mass was categorized into four cases according to its
origin and pathway: two affected continents including South America and the Antarctic Peninsula and
two affected marine cases including the Weddell and Bellingshausen Sea (Fig. 1).
**3. Results and discussion**
**3.1 Characteristics of the NPF events**



### 3.1.1 Occurrence frequency and FR of NPF events

2  After data screening as mentioned in the previous section, 1655-days of data recorded during the

3  observation periods from March 2009 to December 2016 were analyzed. The data including valid data

4  were classified into two groups, NPF event days and non-event days, by using $CN_{2.5-10}$ concentrations

5  measured by two CPCs. The duration of the NPF ranged from 0.6 to 14.4 h, with a mean of $4.6 \pm 1.5$

6  h. Only 6.1% (101 days) of the results were defined as NPF events, whereas 93.9% (1554 days) were

7  classified as the non-NPF events (Table 2). This NPF frequency at King Sejong Station in the Antarctic

8  Peninsula is quite low compared with those in previous studies at other mid-latitude sites (Kulmala et

9  al., 2004; Dal Maso et al., 2005; Pierce et al., 2014; Rose et al., 2015); comparison with other sites in

10  the Antarctic is difficult owing to the lack of long-term observed results. In addition, the monthly

11  variation of the NPF frequency was compared as shown in Fig. 2. It is clear that the NPF number was

12  highest during the austral summer, from December to February, whereas non-events were observed in

13  the austral winter period from June to August. Approximately 72% of the NPF occurred during the

14  summer period, showing the highest value of 38% in January. The clear difference in the frequency of

15  the NPF events in austral summer and winter periods should speculates that solar intensity and

16  temperature play important roles in the formation and growth of aerosol particles, along with precursor

17  vapors derived from marine biota activities in the Antarctic (Virkkula et al., 2009; Kyrö et al., 2013;

18  Weller et al., 2015; Jang et al., 2018).

19  The FR of particles ranging from 2.5 nm to 10 nm varied from 0.16 to 9.88 $cm^{-3}$ $s^{-1}$, with an average

20  of $2.79 \pm 1.05$ $cm^{-3}$ $s^{-1}$. Fig.3 shows the monthly variations in the FR over whole observation periods.

21  The seasonal trend in the FR shows a pattern similar to that of the NPF events frequency. The FRs

22  were the highest during the austral summer (December-February, $3.20 \pm 1.09$ $cm^{-3}$ $s^{-1}$). Those in the

23  austral autumn period (March-May, $1.71 \pm 0.56$ $cm^{-3}$ $s^{-1}$) were similar to those of the spring period

24  (September-November, $1.71 \pm 0.79$ $cm^{-3}$ $s^{-1}$). Because the NPF was observed only one case in May,

25  the result of the May data was ignored in this analysis. In particular, the monthly maximum FR in



December and the minimum in October were 3.52 cm$^{-3}$ s$^{-1}$ and 0.84 cm$^{-3}$ s$^{-1}$, respectively. The FR
measured at various stations in the Antarctic and other continents are summarized in Table 3. The
average level of the FR observed in this study was more than 10 times higher than that of other stations
in Antarctica. Although it is difficult to directly explain the causes of the higher FR, it is likely that the
method used in this study to derive the FR influenced the results. The FRs were estimated in the
previous studies on the basis of the size distribution data with few minute time resolution, whereas the
FR in this study was calculated by using the variation in total number concentration (CN$_{2.5-10}$) data
with a time resolution of 1 s. Another possible reason is the location. As shown in Table 3, the FR at a
coastal region, specifically Mace Head located approximately 500 m from the coast, is higher than that
reported at other sites due to the high biological activity of marine algae, which produce gaseous
precursors from tidal zone and open oceans. Previous modeling research showed that the dimethyl
sulfide emission in the Antarctic Peninsula during the astral summer period is higher than that in other
regions in Antarctica (Yu and Luo, 2010). Thus, the characteristics of the sampling site might have
caused the FR to be higher than that at other site in Antarctica.
**3.1.2 Calculation of other parameters based on size distribution data**

17        On the basis of the size distribution results measured with a SMPS, NPF events were categorized

into three NPF types, as mentioned as Sect. 2.2.2. Type C (which is undefined days) was dominant, as
shown in Table 4; among all NPF event days, only two days (2.0%) were considered as Type A events.
The GRs of nucleation mode particles ranged between 0.02 nm h$^{-1}$ and 3.09 nm h$^{-1}$, with a mean of
0.68 ± 0.27 nm h$^{-1}$. Fig. 4(a) presents the monthly variation in the GR from March 2009 to December
2016. A seasonal trend in the GR is apparent, in which the maximum occurred in the summer. The GR
gradually began to decrease in February whereas and increase again in November, as shown in Fig.
4(a). The GR in January was 0.76 ± 0.26 nm h$^{-1}$, whereas that in November was 0.40 ± 0.15 nm h$^{-1}$.
The GR in this study is similar to the values reported in previous studies conducted in Antarctica. For



instance, Weller et al. (2015) reported that the GR at the Neumayer station varied between 0.4 and 1.9
nm h$^{-1}$, with an average of 0.90±0.46 nm h$^{-1}$. However, our results are lower than those reported by
Järvinen et al. (2013), who studied NPF events at Concordia station, Dome C from December 2007 to
November 2009 and showed a GR of 4.3 nm h$^{-1}$. This discrepancy is likely attributed to the number of
analyzed days. In the present study, we analyzed 86 of 101 NPF days, whereas the previous study
analyzed 15 NPF days.

7        Fig. 4(b) shows a monthly variation in CS during NPF events. The CS varied from $0.02 \times 10^{-3}$ s$^{-1}$

to $25.66 \times 10^{-3}$ s$^{-1}$, with an average of $(6.04 \pm 2.74) \times 10^{-3}$ s$^{-1}$. The value was high in February $((8.17 \pm$
$3.55) \times 10^{-3}$ s$^{-1}$) and a low in April $((2.44 \pm 0.70) \times 10^{-3}$ s$^{-1}$), as shown in Fig. 4(b). The CS measured
in this study was approximately 5-10 times higher than that observed at the other Antarctic station.
Weller et al. (2015), who estimated the CS using light scattering data measured from Neumayer station,
indicated a CS value of about $10^{-3}$ s$^{-1}$. A median CS value of $4.0 \times 10^{-4}$ s$^{-1}$ in a 47-day observation period
at Aboa station was reported by Kyrö et al. (2013). Järvinen et al. (2013) also showed a CS value of
$1.8 \times 10^{-4}$ s$^{-1}$ using data of 15 days.

15       The monthly variation in the condensable vapor source rate during an NPF event is displayed in

Fig. 4(c). The source rates derived were between $0.03 \times 10^{3}$ and $3.74 \times 10^{5}$ cm$^{-3}$ s$^{-1}$, with a mean source
rate of $(5.19 \pm 3.51) \times 10^{4}$ cm$^{-3}$ s$^{-1}$. The source rate of condensable vapor was maximum during the
austral summer months. In particular, the maximum and minimum average values of the source rate
were $(6.40 \pm 3.43) \times 10^{4}$ cm$^{-3}$ s$^{-1}$ in January and $(1.93 \pm 0.92) \times 10^{4}$ cm$^{-3}$ s$^{-1}$ in November, respectively.
This source rate was higher than that measured at a coastal Antarctic station. Kulmala et al. (2005)
reported that the value of source rate varied from $0.9 \times 10^{3}$ cm$^{-3}$ s$^{-1}$ to $2.0 \times 10^{4}$ cm$^{-3}$ s$^{-1}$ at the Aboa station.
**3.3 CCN concentration during NPF events**

24       In this section, the contribution of particle formation to the variation in CCN concentration is

investigated. Although recent studies reported that number concentrations of climate-relevant particles



increased during NPF events (Pierce et al., 2014; Shen et al., 2016; Rose et al., 2017), the contribution
of NPF to CCN concentration was estimated by using an indirect method. The number concentrations
of particles larger than 50, 80 and 100 nm were estimated by using size distribution data. That value
was considered as potential CCN concentration at different supersaturation value. In this study,
however, CCN concentrations at a supersaturation value of 0.4% were measured. Fig. 5 shows
variation in normalized values of $CN_{2.5-10}$ and CCN concentrations as a function of time. The
normalized value was calculated from $CN_{2.5}$ and the CCN concentration at each time divided by the
concentration recorded 1 h prior to the NPF event. The zero in the x-axis in the figure represents the
start time of the NPF event. Data for only 34 days out of 101 NPF days were valid due to the CCN
data availability limited by a malfunctioning of an instrument. The $CN_{2.5-10}$ concentrations sharply
increased at NPF start time and the peak concentration occurred 2 h afterward, as shown in Fig. 5.
Moreover, the CCN concentrations gradually increased for 9 h. Indeed, the maximum CCN
concentrations rose from $170.7\pm38.6\,cm^{-3}$ to $185.6\pm44.6\ cm^{-3}$ during and after the NPF events,
respectively, showing an increase of 9%.
**3.4 Effects of air mass origin on NPF events**

17       The effects of air mass origin on the NPF characteristics were also investigated by 48-h air mass

back trajectory analysis. The frequencies of NPF, FR, GR, CS, and the source rate of condensable
vapor over the whole observation period are listed in Table 5. Here, the analysis results of the NPF
characteristics of air masses originating from South America (Case I) and in an undefined case are not
shown owing to low frequencies. The air masses originating from the sea (Case II and IV) were
dominant during NPF event at King Sejong Station. The FRs were analogous regardless of the air mass
origin and pathway, while the GR of Case III and Case IV was significantly higher than those of Case
II. The lower GR should be related to the CS and the source rate of condensable vapor. In the case of
the air mass originating from the Weddell Sea (Case II), the CS was higher than that of other cases,

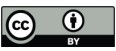


whereas the source rate of condensing vapor was lowest. The higher CS and lower source rate might
a indicate decline in condensing vapor and hence a decrease in GR. Our results for the source rate of
condensable vapor agree with those of a previous study by Yu and Luo (2010), discussed the role of
dimethyl sulfide (DMS) emission in the NPF process in remote oceans. In their simulation study, the
concentrations of DMS and sulfuric acid in the Bellingshausen Sea and the Antarctic Peninsula area
during the austral summer season were higher than those in Weddell Sea region. In satellite-derived
estimates of the biological characteristics, DMS produced from phytoplankton was found to be more
dominant in the Bellingshausen Sea than in the Weddell Sea (Jang et al., 2018). Sulfuric acid is derived
from oxidation of DMS in oceans (Virkkula et al., 2009). In this study, the condensable vapor was
assumed to be sulfuric acid in the source rate calculations, as mentioned in Sect. 2.2.3.

11       Fig. 6 shows a comparison of the NPF characteristics depending on the origin and pathway of the

air mass during the summer season. The mean CS value was high. However, in case of the air mass
originating from the Bellingshausen Sea (Case IV), the GR was relatively higher than the values of air
masses originated from other region due to the higher values of the source rate of condensable vapor.
The mean value of this source rate for the air mass originating from the Weddell Sea (Case II) was
similar to that from the Antarctic Peninsula (Case III), while the CS mean value was 1.7 times higher.
This resulted in a low GR.

18       For air mass originating from the Bellingshausen Sea (Case IV), the seasonal properties of the

parameters related to the NPF events were analyzed. As shown in Fig. 7, the mean values of FR, GR
and the source rate of condensable vapor were highest during the austral summer periods. However,
mean values of CS were highest during the spring period.
**4. Summary**

24       In this study, the characteristics of NPF at King Sejong station in Antarctic Peninsula were

investigated using a data set of eight years from March 2009 to December 2016, of total particle



number concentrations and particle size distributions. The frequencies of NPF events and FR were
obtained by using the data of total number concentrations, whereas GR, CS and the source rate of
condensable vapor were calculated from the aerosol size distribution results. A low occurrence
frequency of NPF events, at 6%, was observed, and most of the NPF events occurred during the austral
summer. No NPF events were observed during the winter due to lower solar radiation and a lack of
precursors for particle formation. The mean values of the FR and GR were $2.79 \pm 1.05$ cm$^{-3}$ s$^{-1}$ and
$0.68 \pm 0.27$ nm h$^{-1}$, respectively. These results show that the FR at King Sejong Station as higher than
that at other Antarctica sites, whereas the GR was relatively similar to values reported in previous
studies conducted in the Antarctic. A possible reason for the lower GR can be attributed to the CS,
which was 5-10 times higher than that reported at other stations in Antarctica. This observation
suggests that condensable vapor contributed to growth of nucleated nanoparticles and may have
condensed onto pre-existing particles, hence decreasing the GR. According to 48-h backward
trajectory analysis, air masses originating from oceanic areas were dominant during the NPF events.
In order to investigate the contribution of the NPF events to variation in CCN concentrations at a
supersaturation value of 0.4%, the CCN concentrations were compared with the CN$_{2.5-10}$
concentrations as a function of time. The results showed that the CCN concentrations during and after
the NPF events increased approximately 9% compared with those measured before the event. This
study is the first to report the characteristics of NPF in the Antarctic Peninsula. However, further
research is need to understand the chemical characteristics of aerosol particles and the chemical
composition of precursors during NPF events to fully understand the NPF for this region.
**Author contributions**
JK and YJY designed the study, YG, JHC, HJK, KTP, JP, and BYL analysed aerosol data. JK and
YJY prepared the manuscript with contributions from all co-authors.
**Acknowledgements**



We would like to thank the many technicians and scientists of the overwintering crews. This work was
supported by the KOPRI project (PE18010) and a Korea Grant from the Korean Government (MSIP)
(NRF-2016M1A5A1901769) (KOPRI-PN18081).

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



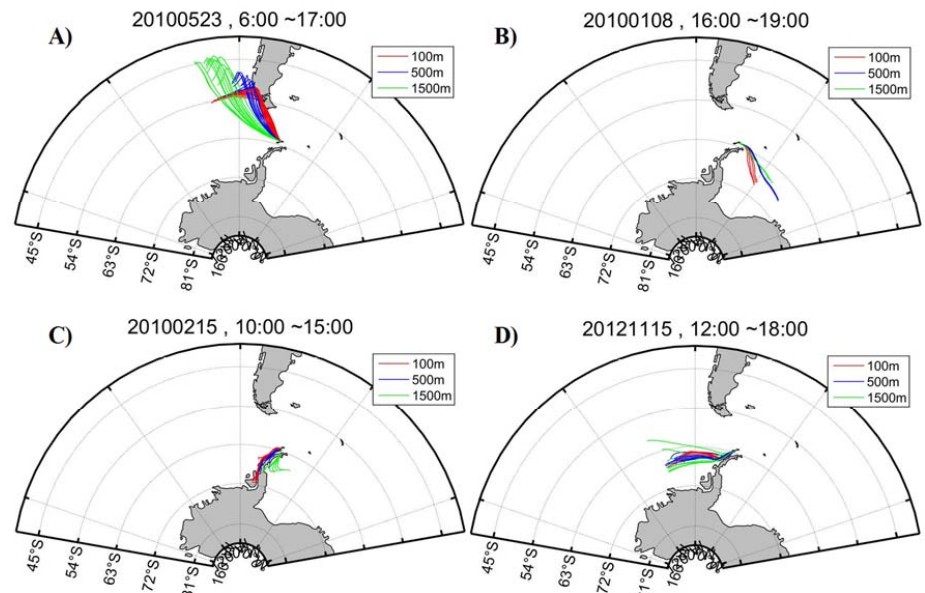

Figure 1. Example of the four cases considering to the air mass origin and pathway: (a) South
America, (b) Weddell Sea, (c) Antarctic Peninsula, and (d) Bellingshausen Sea. Typical 48-h air mass
backward trajectories were analyzed, ending at heights of 100m (Red line), 500m (Blue line) and
1500m (Green line) above the ground level of the sampling site.



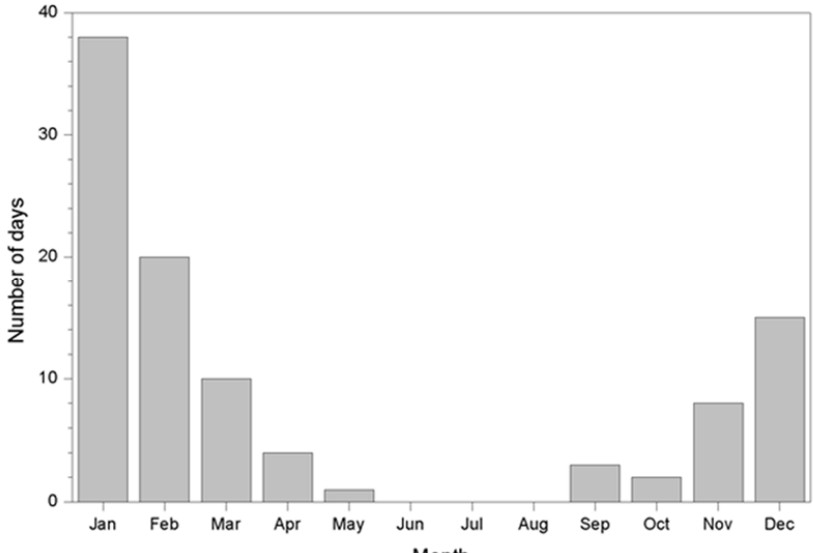

4    Figure 2. Monthly variation in the number of NPF days between March 2009 and December 2016.



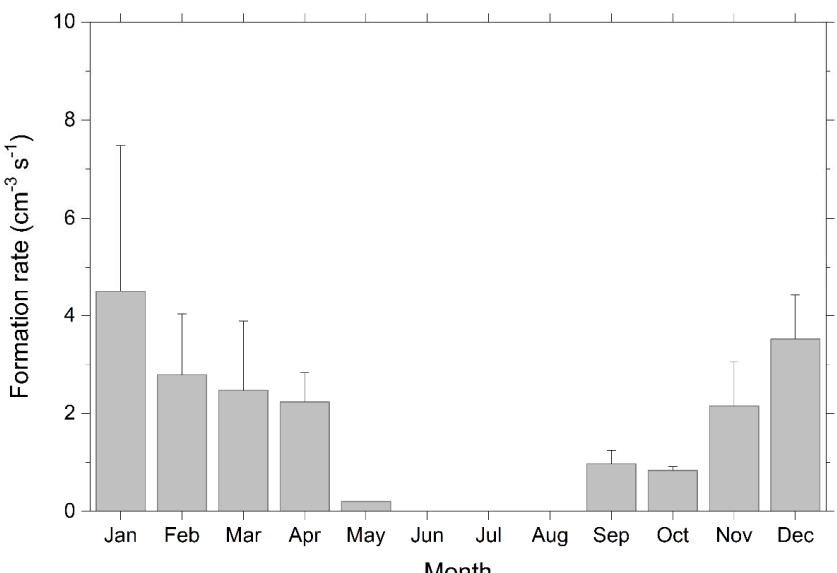

4    Figure 3. Monthly variation in the formation rate. The error bars represent standard deviation





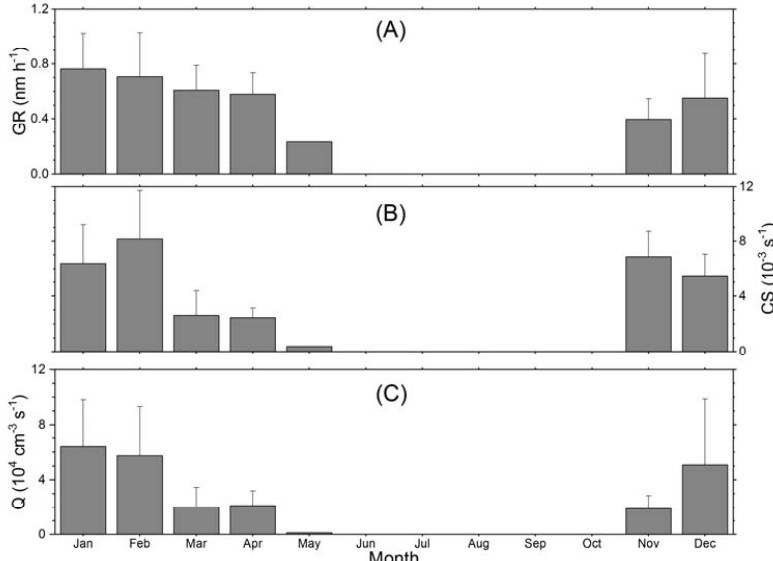

4 Figure 4. Monthly variation in (a) the growth rates (GR) of nucleation mode particles ranging from 10

5 nm to 25 nm, (b) the condensation sink (CS), and (c) the source rate of condensable vapor (Q). The

6 error bars represent standard deviation.





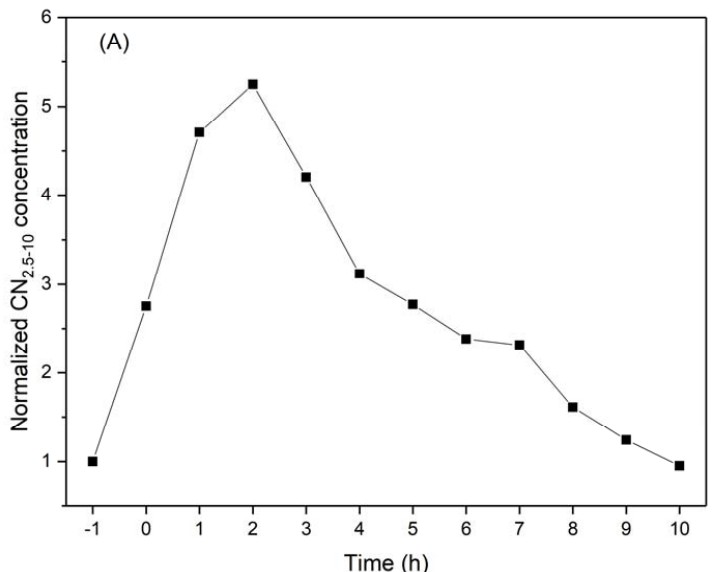

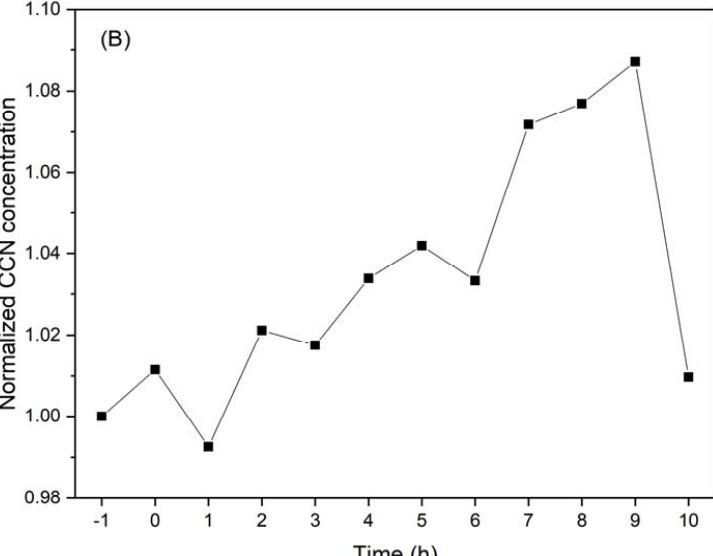

4    Figure 5. Variation in normalized (a) $CN_{2.5-10}$ and (b) CCN concentration with time. The zero in the x-

5    axis indicates the start time of the NPF events.



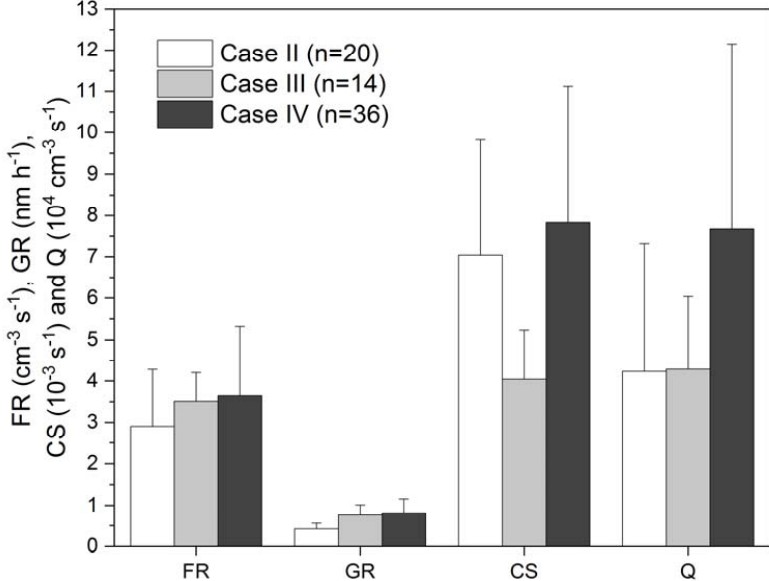

Figure 6. Comparison of NPF characteristics including the formation rate (FR), growth rate (GR), condensation sink (CS) and source rate of condensable vapors (Q) depending on the origins and pathway of air masses during the astral summer period. The error bars represent standard deviation.





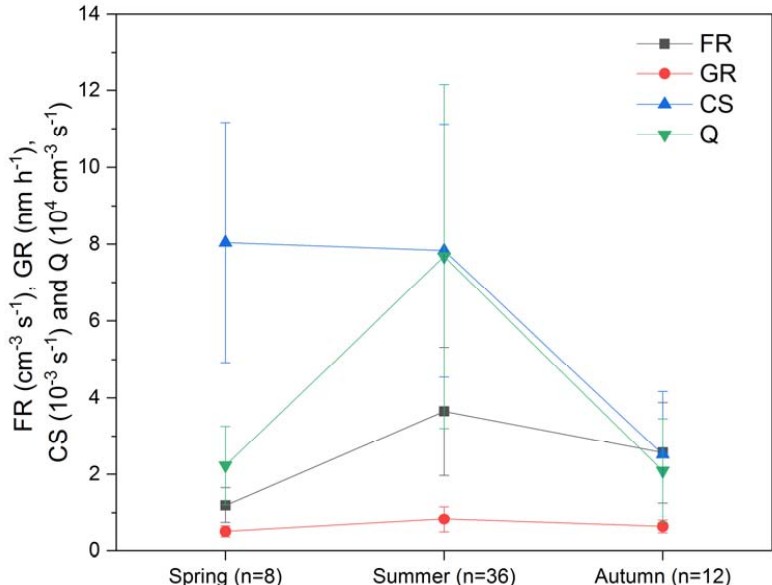

Figure 7. Seasonal characteristics of parameters related to NPF events in which the air masses originated from the Bellingshausen Sea. FR, GR, CS, and Q refer to formation rate, growth rate, condensation sink, and source rate of condensing vapor, respectively. The error bars represent standard deviation.





2 Table 1. Summary of data acquisition rate for each instrument during the analysis periods

| Measurement parameter | Instrument | Data acquisition rate(%) |
| --- | --- | --- |
| Number concentration of particle larger than 2.5 nm | CPC (TSI 3776) | 80.7 |
| Number concentration of particle larger than 10 nm | CPC (TSI 3772) | 79.5 |
| Size distribution | SMPS | 40.3 |
| CCN concentrations | CCNC | 36.4 |

4

6 Table 2. Event statistics classified by using total concentration data obtained from two CPCs

| | Days | Percentage of total days |
| --- | --- | --- |
| NPF events | 101 | 6.1 |
| Non events | 1554 | 93.9 |
| Total | 1655 | |





Table 3. Summary of the formation rates observed at different sampling site in Antarctica and in other continents. DMPS, SMPS, and CPC mean differential mobility particle sizer, scanning mobility particle sizer, and condensation particle counter, respectively.

| Site | Period | Method | Formation rates (cm⁻³ s⁻¹) | | References |
|---|---|---|---|---|---|
| King Sejong (Antarctic Peninsula) | 03/2009 ~ 12/2016 | Two CPCs (TSI 3772 & TSI 3776) | 2.79 | $J_{2.5\text{-}10}$ | This study |
| Syowa (Antarctica) | 08/1978 ~ 12/1978 | | $3.8\times10^{-4}$ | $J_{10}$ | Ito, 1993 |
| Dome C (Antarctica) | 12/2007 ~ 11/2009 | DMPS | 0.038 | $J_{10}$ | Järvinen et al., 2013 |
| Aboa (Antarctica) | 01/2010 | DMPS | 0.003 ~ 0.3 | $J_{10}$ | Kyrö et al., 2013 |
| Neumayer (Antarctica) | 20/01/2012 ~ 26/03/2012 01/02/2014 ~ 30/04/2014 | SMPS | 0.02 ~ 0.1 | $J_{3\text{-}25}$ | Weller et al., 2015 |
| Värriö (Sub Arctic) | 12/1997 ~ 07/2001 | DMPS | 0.38 | $J_{10}$ | Dal Maso, 2002 |
| Hyytiälä (Rural) | 1996 ~ 2003 | DMPS | 0.61 | $J_{3\text{-}25}$ | Dal Maso et al., 2005 |
| Mace Head (Coastal) | 1996 ~ 1997 | Two CPCs (TSI 3022 & TSI 3025) | $10^2 \sim 10^4$ | $J_{3\text{-}10}$ | Grenfell et al., 1999 |
| Jungfraujoch (Remote) | 03/1997 ~ 05/1998 | SMPS | 0.14 | $J_{10}$ | Weingartner et al., 1999 |
| Dresden area (Rural) | 1996 ~ 1998 | Two CPCs (UCPC & CPC) | 110 | $J_{10}$ | Keil and Wendisch, 2001 |
| Atlanta (Urban) | 08/1998 ~ 08/1999 | Nano-SMPS | 10 ~ 15 | $J_3$ | Woo et al., 2001 |
| Shangdianzi (Rural) | 03/2008 ~ 12/2013 | DMPS | 6.3 | $J_3$ | Shen et al., 2016 |

2





Table 4. NPF event classification statistics using size distribution results. Type A refers to days in which
the formation and growth of particles were clear. Type B refer to days in which the formation occurred
but the growth was not clear. Type C refers to days in which the event occurrence was unclear.

|        | Days | Percentage of NPF days |
|--------|------|------------------------|
| Type A | 2    | 2.0                    |
| Type B | 37   | 36.6                   |
| Type C | 62   | 61.4                   |
| Total  | 101  |                        |

Table 5. Summary of NPF characteristic statics depending on the air mass origin. FR is the formation
rate, GR is the growth rate, CS is the condensation sink, and Q is the source rate of condensable vapor.
Case I, Case II, Case III, and Case IV refer to the origin and pathway of air masses from South America,
the Weddell Sea, the Antarctic Peninsula, and the Bellingshausen Sea, respectively.

|           | NPF days | FR $(cm^{-3} s^{-1})$ | GR $(nm\ h^{-1})$ | CS $(10^{-3} s^{-1})$ | Q $(10^4\ cm^{-3} s^{-1})$ |
|-----------|----------|------------------|------------------|------------------|------------------|
| Case I    | 3        |                  |                  |                  |                  |
| Case II   | 24       | $2.81 \pm 1.29$  | $0.41 \pm 0.15$  | $6.95 \pm 2.65$  | $3.87 \pm 2.90$  |
| Case III  | 16       | $3.10 \pm 0.80$  | $0.77 \pm 0.25$  | $4.19 \pm 1.30$  | $4.29 \pm 1.75$  |
| Case IV   | 56       | $3.08 \pm 1.55$  | $0.76 \pm 0.30$  | $6.79 \pm 3.20$  | $6.20 \pm 4.08$  |
| Undefined | 2        |                  |                  |                  |                  |

