# Peer review of "New particle formation observed at King Sejong Station, Antarctic"

_Atmospheric Chemistry and Physics, 2018_

## Referee Comment (RC1) · Anonymous Referee #1 · 18 Dec 2018

General comments:

The manuscript at hand characterizes new particle formation (NPF) events observed at the Korean Antarctic Station King Sejong. As the authors state, this is the first NPF investigation from the Antarctic Peninsula. To my knowledge it is based on the longest observation period actually measured in Antarctica regarding this topic. In addition, the authors discussed in particular NPF events along with cloud condensation nuclei (CCN) data. The article is written in a straight and concise way and presents invaluable results to elucidate NPF and its impact on CCN availability at this site and (coastal)

[Figure]

Antarctica in general.

However, the article in the present appearance has some weak points. Especially the regrettably scarce discussion in general is not commensurate with the unquestionable valuable data set. In addition, description of the used instruments and data evaluation procedures are often insufficient (see specific comments below). I think this outstanding data set is worth the effort addressing this weakness and considering a more in-depth discussion.

Notwithstanding, I am confident that the data and their evaluation presented here are of high quality and on the whole, the subject is appropriate to ACP. Hence, I recommend accepting the paper after revisions according to my specified suggestions from above and listed below.

Specific comments:

1. The authors should provide specification and operation details for the SMPS and CCN instruments even though they were comparable to Kim et al. (2017). Furthermore, I miss an adequate presentation of the SMPS results! In any case, it would be advisable to show some figures (e.g. the typical Dp vs. time contour plots), at least for the two case "A" NPF events.

2. Another concern is the lack of an appropriate CCN data presentation. For instance, the authors showed no figures regarding the CCN spectrum. I would appreciate a thorough description of the performed data analysis.

Chapters 2.2.1. and 2.2.2: The authors based their definition and classification of NPF events on the criteria compiled by Dal Maso et al. (2005) and Yli-Juuti et al. (2009), which are widely accepted by the community. According to these previous studies, an NPF event must show signs of growth (see Dal Maso et al., 2005, p. 326). Therefore, NPF events can only be identified by size distribution (here SMPS) data, but clearly not by sole CN2.5 minus CN10 data. The latter just indicate a potential NPF, which may be

better termed as particle burst.

Chapters 2.2.3: I guess that CN and especially CN2.5 – CN10 concentration data based on 1 s resolution are highly fluctuating, making the FR evaluation somewhat arbitrary. Please specify in more detail the way you extract dNnuc/dt from the data (maybe by showing a representative figure?).

Chapters 2.3: Please specify, in which way/procedure you have characterized air masses (by cluster analysis or just "manually")?

Results and Discussion chapter and Tables 4 and 5: The authors observed just two type "A" NPF events, from which growth can be determined with confidence according to Dal Maso et al. (2005). For type "B" events, the authors state that growth was not clear (see caption of table 4). I am confused about this: Does this mean the bottom line is that the reported growth rates were based on merely two events? Please clarify this point!

Chapter 3.3: From my point of view the presented discussion is inadequate. Evaluation of the CCNC data demands a more detailed description and discussion. Especially: A systematic analysis along with SMPS data would be crucial and should be presented. Are your CCNC results consistent with SMPS data?

Chapter 3.4, lines 12 through 14: The authors argue that higher GR observed in air masses emerging from the Bellinghausen Sea area are due to higher source rates of condensable vapour. Unless I am very much mistaken, this is a typical case of circular reasoning, because regarding eqs. (4) and (5), the source rate Q is linearly dependent on the GR, isn't it!

Technical corrections:

Page 1 (abstract), line 16: Misleading phrase. Change to "…during particle bursts (not during a particle burst)…"

Page 1 (abstract), lines 23, 27, and throughout the text: Please present measured

values and values derived from data just with their relevant/meaningful digits.

Page 3, line 3: Misleading phrase. Change to "A NPF event occurring in the period between December 1998 and December 2000. . ."

Page 4, line 15: Modify to ". . .raw data measured during the following conditions. . ."

Page 4, line 21 and throughout the manuscript: Delete "value" in "value difference".

Page 5, line 24: Use a unique consistent term for particle number concentrations between 2.5 nm and 10 nm (either CN2.5-10 or CN2.5 – CN10) throughout the text!

Page 6, line 15: "speed" should be "loss rate".

Page 8, line 15: "should speculate" should be "indicates".

Page 9, line 18: Delete "(which is undefined days)".

Page 9, line 23: Delete "whereas".

Page 11, line 18-21: I do not understand the meaning of this phrase - please clarify: What is meant with "undefined case" here? Delete this line in Table 5.

Page 12, line 2: ". . .a indicate decline. . ." should be ". . .indicate a decline. . ."

Page 12, line 3: ". . .discussed. . ." should be ". . .discussing. . ."

Page 12, line 4: ". . .simulation. . ." should be ". . .model. . ."

Page 12, line 7: Misleading phrase. The term "estimates of the biological characteristics" is somewhat vague, please specify.

Page 12, lines 8-9: DMS oxidation to sulphuric acid occurs in the atmosphere but not in the ocean – please correct!

---

## Referee Comment (RC2) · Anonymous Referee #2 · 9 Jan 2019

Review comment on "New particle formation events observed at King Sejong Station, Antarctic Peninsula – Part 1: Physical characteristics and contribution to cloud condensation nuclei" by Jaeseok Kim et al.

This manuscript presents new particle formation (NPF) and its impact on CCN ability at Korean Antarctic research Station (King Sejong) located in the Antarctic Peninsula. This study is based on long-term aerosol measurements for several years. To our knowledge, the long-term SMPS measurements through the years in the Antarctic regions are very limited. Actually, results in the manuscript are important and interesting

to understand NPF and aerosol science in the Antarctic regions. As a whole, the topic of the manuscript is relevant and suitable for the scope of the "Atmos. Chem. Phys.". However, there are several points which require some careful revision and corrections before publication.

Major points

1. Authors showed NPF occurrence and frequency in Section of 3.1.1. However, time series of CN concentrations and SMPS results (i.e. contour plots of variations of aerosol size distributions) should be shown and add explanation before analysis/discussion of NPF occurrence and frequency. The plots of the typical examples can provide important information for us.

2. It is true that emission of aerosol precursors from oceanic bioactivity and atmospheric photochemical reactions are associated with NPF in the Antarctic coasts during summer. Unlike to other Antarctic coastal regions, however, anthropogenic impacts (local contamination) can be larger around the Antarctic Peninsula particularly in the summer because of activity in many stations and ship-borne tourism. Therefore, influence of anthropogenic activity and local contamination should be analyzed and discussed before discussion on contribution of condensable vapors originated from oceanic bioactivity. The following works are useful references.

Shirsat, S. V. and Graf, H. F.: An emission inventory of sulfur from anthropogenic sources in Antarctica, Atmospheric Chemistry and Physics, 9(10), 3397–3408, 2009.

Graf, H.-F., Shirsat, S. V., Oppenheimer, C., Jarvis, M. J., Podzun, R., and Jacob, D.: Continental scale Antarctic deposition of sulphur and black carbon from anthropogenic and volcanic sources, Atmospheric Chemistry and Physics, doi:10.5194/acp-10-2457-2010, 2010.

Furthermore, air masses in the Antarctic Peninsula were transported frequently from south America. This transport pathway can lead to high aerosol number concentrations

and BC concentrations at Ferraz Station located in the Antarctic Peninsula (Pereira et al., 2004, 2006). In other words, these studies implied that anthropogenic aerosol precursors and land-origin aerosol precursors such as organics can be transported and supplied to the Antarctic Peninsula. Thus, I recommend strongly comparison of number concentrations, NPF frequency, and FR in each air mass origin.

Pereira, K., Evanhelista, H., Pereira, E., Simões, J., Johnson, E., and Melo, L.: Transport of crustal microparticles from Chilean Patagonia to the Antarctic Peninsula by SEM‐EDS analysis, Tellus B, 56(3), 262–275, doi:10.1111/j.1600-0889.2004.00105.x, 2004.

Pereira, E., Evangelista, H., Pereira, K., Cavalcanti, I., and Setzer, A.: Apportionment of black carbon in the South Shetland Islands, Antarctic Peninsula, Journal of Geophysical Research: Atmospheres (1984–2012), 111(D3), doi:10.1029/2005JD006086, 2006.

3. Authors stated definition and classification of NPF in Section of 2.2.1 and 2.2.2. Because SMPS measured size distributions of aerosol particles with size range of D>10 nm, authors tried likely to identify NPF using the difference of CN concentrations (e.g., CN2.5-CN10). Criteria values of 500 cm-3 were used for the NPF identification. What is the procedure to decide the criteria values? This criteria is very important basic in this study. I think that authors were in accordance of procedures shown by Humphrires et al. (2016). Considering that measuring site and conditions were different to sea-ice area (Humphrires et al., 2016), authors should show example plots of time series of CN2.5-CN10 and discuss the suitable criteria values. In addition, classification of NPF in accordance with previous works (Dal Maso et al., 2005; Yli-Juuti et al., 2009) requires information about particle growth after NPF. However, the difference of CN concentrations cannot provide information on particle growth. How did you identify particle growth of nucleation mode (D<10 nm)?

4. FR was estimated from CN data with 1 sec resolution in this study. What is values

of variability? In general, 1 sec CN data can be varied greatly. The large variability engender the large error of the estimated FR. CN data with longer time resolution (e.g., one minute) is better to estimate FR. Also, statistical analysis and error estimation are required for CN2.5-CN10 and the estimated FR.

5. GR was estimated from GMD. How did you calculate GMD? Did you have log-normal fitting analysis or identify diameter of mode maximum? Some explanation is needed in Section of 2.2.3.

6. To identify origins and pathway of air masses with NPF, some trajectory was shown in Figure 1. Although trajectory can provide us important information of transport processes of air masses, Fig.1 showed only some cases. I suggest all trajectories in NPF at each height are plotted in Figure 1 (e.g., trajectory density map) to identify origins and pathway of air masses with NPF.

7. CCN concentrations were discussed in Section of 3.3. Long-term CCN records provide important knowledge to us. In this study, aerosol size distributions were measured simultaneously by SMPS. Nevertheless, aerosol size distributions did not compare to CCN data. I understand that critical diameter was estimated hardly in this study. However, aerosol size distributions must be useful and important data to elucidate features of CCN concentrations. The critical diameter of the Antarctic aerosols during summer was discussed by Kyrö et al. (2013). Comparison between size distributions and CCN should be shown and discussed.

Minor points

1. Introduction: Page 2 Line 20 Aerosol particles with size larger than several tens nm are not "new".

2. Introduction: Page 3 Line 3

"Dall'osto" is correct.

3. Introduction: Page 3 Line 10-12

Asmi et al. (2010) presented hygroscopicity of ultrafine particles measured at the coastal Antarctic station (Aboa). They showed and discussed hygroscopic growth factor and CCN activity, although they did not measure directly CCN. This should be mentioned in introduction.

4. Section 2.1 Measuring periods should be mentioned in Methods section, although the periods was shown in the section of Results and discussion.

5. Page 8 Line 14-18 Kyrö et al. (2013) showed emission of aerosol precursors from melt pond, not from oceanic bio-activity. This description should be modified.

"∼biota activities in the Antarctica" is correct.

6. Page 8 line 24-25 In this study, the NPF was observed in May in spite of only one case. If NPF occurred actually in the Antarctica in May, this is important to understand aerosol science in the Antarctic troposphere. Some explanation and discussion such as FR and air mass origin should be added.

7. Figures 3 and 4 Both figures can be merged. That is easy to compare among each other.

8. Page 9 Line 22-24 GRs in September-October were not shown in Fig. 4. Does it mean no particle growth in September-October? Some explanation should be added.

9. Page 10 Line 9-10 Higher CS values were obtained at King Sejong Station. The high CS might result from high aerosol number concentrations, although high CN related also to aerosol size distributions. If so (high aerosol concentrations), supply and transport of aerosols and aerosol precursors should be taken into account. This must be associated with FR, GR, and CCN ability. Details were already shown in the major comment.

10. Page 11 Line 19-22 Are air mass origins (Case I-IV) corresponding to Fig.1a-d?

11. Section of 3.4 CS values were used for discussion. I suggest that CS values and

aerosol number concentrations obtained in previous works at stations (e.g., Neumayer and Aboa) around Weddle Sea should be compared to data in this study. As mentioned in major comment, anthropogenic and local impact should be discussed. Such impacts are analyzed hardly only by trajectory.

———————————————

---

## Author Comment (AC1) · 10 Apr 2019

We thank Referee 1 for providing insightful suggestions that have considerably improved the readability of the revised manuscript. Our responses to general and specific comments raised by Referee 1 are stated below. The revised manuscript was uploaded in the form of a supplement

General comments:

The manuscript at hand characterizes new particle formation (NPF) events observed at the Korean Antarctic Station King Sejong. As the authors state, this is the first NPF investigation from the Antarctic Peninsula. To my knowledge it is based on the longest observation period actually measured in Antarctica regarding this topic. In addition, the authors discussed in particular NPF events along with cloud condensation nuclei (CCN) data. The article is written in a straight and concise way and presents invaluable results to elucidate NPF and its impact on CCN availability at this site and (coastal) Antarctica in general. However, the article in the present appearance has some weak points. Especially the regrettably scarce discussion in general is not commensurate with the unquestionable valuable data set. In addition, description of the used instruments and data evaluation procedures are often insufficient (see specific comments below). I think this outstanding data set is worth the effort addressing this weakness and considering a more in-depth discussion. Notwithstanding, I am confident that the data and their evaluation presented here are of high quality and on the whole, the subject is appropriate to ACP. Hence, I recommend accepting the paper after revisions according to my specified suggestions from above and listed below.

Specific comments:

1. The authors should provide specification and operation details for the SMPS and CCN instruments even though they were comparable to Kim et al. (2017). Furthermore, I miss an adequate presentation of the SMPS results! In any case, it would be advisable to show some figures (e.g. the typical Dp vs. time contour plots), at least for the two case "A" NPF events.

Authors' response: We have described the specification and operation details for the SMPS and CCN system in our previous paper (Kim et al., 2017). Following referee's advice, in the revised manuscript, we modified the paragraph on Page 4 Line 5 to clarify the specification and operation details for the SMPS and CCN instruments.

*"The aerosol size distributions of particles ranging from 10 to 300 nm were measured every 3 minutes with a scanning mobility particle sizer (SMPS) consisting of a differential mobility analyzer (DMA; HCT Inc., LDMA 4210) and a CPC (TSI 3772). The flow rate of sheath air and aerosol flow of DMA were 10 L min$^{-1}$ and 1 L min$^{-1}$, respectively. The CCN concentrations were simultaneously measured by using a CCN counter (DMT CCN-100) with five different supersaturation values (i.e. 0.2, 0.4, 0.6, 0.8 and 1.0%). The sampling duration was set to be 5 minutes for each supersaturation value (except for 0.2%). For the 0.2% supersaturation value, the CCN concentration was measured for 10 min because of stability after measurements at 1% supersaturation value. In the present work, only results of CCN concentration for a 0.4% supersaturation value were used."*

We also added following contour figures in the revised manuscript.

[Figure]

Figure 1. Example of types of the NPF based on the SMPS data. (a) type A (18 January 2011-20 January 2011), (b) type B (13 January 2015) and (c) type C (9 January 2015). Type A is days when the formation and growth of nanoparticles should be clear. Type B is days when the formation occurred but growth was not clear. Type C is days when it cannot be said whether there is an event or not.

2. Another concern is the lack of an appropriate CCN data presentation. For instance, the authors showed no figures regarding the CCN spectrum. I would appreciate a thorough description of the performed data analysis.

Authors' response: CCN data were obtained at five different supersaturation ratio values (0.2, 0.4, 0.6, 0.8 and 1.0%) using commercial CCNC (DMT CCN-100). The sampling time was set at 5 min including stability duration for each supersaturation value except for 0.2% supersaturation value. For 0.2% supersaturation value, the CCN data was collected for 10 min because the additional time was needed to achieve stability after measurements at 1.0% supersaturation value. Based on previous study (Anttila et al., 2012) which compared relationship between CCN concentration and cloud droplet number concentration, in this study, hourly mean CCN data at 0.4% supersaturation value were used. Based on this results, we compared variation in normalized values of $CN_{2.5-10}$ and CCN concentrations during NPF period as shown in Figure 2.

To explain method for CCN data analysis, we added the following sentence in Page 4 Line 8:

*"The CCN concentrations were simultaneously measured by using a CCN counter (DMT CCN-100) with five different supersaturation values (i.e. 0.2, 0.4, 0.6, 0.8 and 1.0%). The sampling duration was set to be 5 minutes for each supersaturation value (except for 0.2%). For the 0.2% supersaturation value, the CCN concentration was measured for 10 min because of stability after measurements at 1% supersaturation value. In the present work, only results of CCN concentration for a 0.4% supersaturation value were used."*

[Figure]

Figure 2. Example of comparison among CN concentrations from CPC data (upper panel), size distribution from SMPS data (middle panel) and hourly mean CCN concentration (bottom panel) at 0.4% supersaturation value as a function of time on 30 March 2009.

Chapters 2.2.1. and 2.2.2: The authors based their definition and classification of NPF events on the criteria compiled by Dal Maso et al. (2005) and Yli-Juuti et al. (2009), which are widely accepted by the community. According to these previous studies, an NPF event must show signs of growth (see Dal Maso et al., 2005, p. 326). Therefore, NPF events can only be identified by size distribution (here SMPS) data, but clearly not by sole CN2.5 minus CN10 data. The latter just indicate a potential NPF, which may be better termed as particle burst.

Authors' response: According to previous study (Dal Maso et al., 2005), an NPF event must show signs of growth. Authors acknowledge and agree that the SMPS data are widely used for identification and classification of the NPF events. However, in this study, because the availability of SMPS data set was lower than that of CPC data set, CN2.5 and CN10 data were used to define NPF events and SMPS data used to classify types of the NPF events. For identification of the NPF events, $CN_{2.5-10}$ data, $CN_{2.5-10}/CN_{10}$ data and the duration time data were used as mentioned at Section 2.2.1. In particular, $CN_{2.5-10}/CN_{10}$ values can be used to distinguish newly formed particles to background particles (Warren and Seinfeld, 1985; Covert et al., 1992; Humphries et al., 2015). Since we used strict category to define the burst of nanoparticles, this supports the widely-used definition of the NPF events. Authors think the definition of the NPF event is feasible in this study.

Chapters 2.2.3: I guess that CN and especially CN2.5 – CN10 concentration data based on 1 s resolution are highly fluctuating, making the FR evaluation somewhat arbitrary. Please specify in more detail the way you extract dNnuc/dt from the data (maybe by showing a representative figure?).

Authors' response: Authors appreciate the issue raised by the referee. Because CN data with 1 s time resolution are highly fluctuating, FR was estimated using an one-minute averaged CN concentration. To calculate the FR values, we first checked $CN_{2.5-10}/CN_{10}$ values. The $CN_{2.5-10}/CN_{10}$ values can be used to distinguish between newly formed particles and background particles events (Warren and Seinfeld, 1985; Covert et al., 1992; Humphries et al., 2015). As shown in Figure 3, when $CN_{2.5-10}/CN_{10}$ values were higher than 10, we considered as NPF events as mentioned Sec 2.2.1. Time variation (dt) was estimated from the time it starts to increase of $CN_{2.5-10}/CN_{10}$ to the time it was highest values. Variation of CN2.5-10 concentration ($dN_{nuc}$) was calculated at that time.

[Figure]

Figure 3. Example for estimation of the formation rate during NPF event on 7 April 2009: (a) $CN_{2.5-10}/CN_{10}$ and (b) $CN_{2.5-10}$ concentration with 1 min time resolution.

To clarify we modified sentence to following text on Page 6 Line 2:
*"On the basis of the average number concentration data with 1 min time resolution, the FR was calculated for cases in which $CN_{2.5-10}/CN_{10}$ values and $CN_{2.5-10}$ concentrations sharply increased (Fig. S1 in the Supplement)."*

Chapters 2.3: Please specify, in which way/procedure you have characterized air masses (by cluster analysis or just "manually")?

Authors' response: Air mass backward trajectory analysis during the NPF event periods was conducted by using the HYSPLIT model (http://www.arl.noaa.gov/HYSPLIR.php). The origin of air masses arriving at the observation site during the NPF events (a total of 101 event days) was manually categorized into four cases by analyzing 48-h backward trajectory data ending at height of 100m, 500m and 1500 m above the ground level. For instance, if time of the NPF events was from 13:00 to 17:00, we run the 48-h air mass backward trajectory for each hour. The results with similar air mass origins and pathways during the NPF event periods at three different heights were used for the analysis in this study.

To clarify we modified paragraph to following text on Page 8 Line 1:
*"The origin of air masses arriving at the observation site during the NPF events (a total of 101 event days) was manually categorized into four cases by analyzing 48-h backward trajectory data ending at height of 100, 500 and 1500 m above the ground level. The results with similar air mass origins and pathways during the NPF event periods at three different heights were used for the analysis in this study, as shown in Fig. 2"*

Results and Discussion chapter and Tables 4 and 5: The authors observed just two type "A" NPF events, from which growth can be determined with confidence according to Dal Maso et al. (2005). For type "B" events, the authors state that growth was not clear (see caption of table 4). I am confused about this: Does this mean the bottom line is that the reported growth rates were based on merely two events? Please clarify this point!

Authors' response: Aerosol size distribution data were used for classification of the NPF events. Based on the contour plots of aerosol size distribution, type of the NPF events was classified. For the calculation of growth rates, hourly mean aerosol size distribution data was used for all types of NPF. The geometric mean dimeter (GMD) of particles which is limited to the size range of 10-25 nm was used. According to these method, growth rate of particles ranging from 10 to 25 nm was estimated regardless of type of the NPF events as shown in Figure 4.

To clarify we modified paragraph to following text on Page 6 Line 16:
*"Based on the hourly mean aerosol size distribution data, the geometric mean dimeter (GMD) of particles which is limited to the size range of 10-25 nm was used. Here, the GMD was calculated from log-normal fitting analysis. According to these method, growth rate of particles ranging from 10-25 nm was estimated regardless of the NPF event types (Fig. S2 in the Supplement)"*

[Figure]

Figure 4. Geometric mean diameter (GMD) of particles ranging from 10 nm to 25 nm as a function of the time: the growth rate (nm h-1) was calculated as the regression slope. The LST means local standard time.

Chapter 3.3: From my point of view the presented discussion is inadequate. Evaluation of the CCNC data demands a more detailed description and discussion. Especially: A systematic analysis along with SMPS data would be crucial and should be presented. Are your CCNC results consistent with SMPS data?

Authors' response: As described in section 3.3, according to previous studies (Pierce et al., 2014; Shen et al., 2016; Rose et al., 2017), in order to understand relationship between NPF event and CCN concentration, it was suggested that number concentrations of particles larger than 50, 80 and 100 nm estimated by SMPS data are compared with aerosol size distribution data. While, in this study, CCN concentration measured directly by CCN counter were compared concentration of newly formed particles ($CN_{2.5-10}$) as the function of time during NPF event periods. Since it was very rare when the all 3 instruments – CPCs, SMPS, and CCN counter – are running together with the very best condition during the particle burst event, authors decide to choose the best way available, comparing CPC data with CCN during the 34 days with two dataset are available. In this manuscript, authors want to show the results that the CCN concentration increase are noticed for a couple of hours following NPF event under clean Antarctic environment, and this results are derived directly from in-situ CCN measurements.

Chapter 3.4, lines 12 through 14: The authors argue that higher GR observed in air masses emerging from the Bellinghausen Sea area are due to higher source rates of condensable vapour. Unless I am very much mistaken, this is a typical case of circular reasoning, because regarding eqs. (4) and (5), the source rate Q is linearly dependent on the GR, isn't it!

Authors' response: The referee pointed out correctly. To clarify we modified sentence to following text on Page 12 Line 20:

*"However, in case of the air mass originating from the Bellingshausen Sea (Case IV), the GR was relatively higher than the values of air masses originated from other region."*

Technical corrections:

Page 1 (abstract), line 16: Misleading phrase. Change to ": : :during particle bursts (not during a particle burst): : :"

Authors' response: We changed it (Page 1 Line 16).

Page 1 (abstract), lines 23, 27, and throughout the text: Please present measured values and values derived from data just with their relevant/meaningful digits.

Authors' response: Thanks! We modified it (Page 1 Line 23; Page 1 Line 27).

Page 3, line 3: Misleading phrase. Change to "A NPF event occurring in the period between December 1998 and December 2000: : :"

Authors' response: We changed it (Page 3 Line 3).

Page 4, line 15: Modify to ": : :raw data measured during the following conditions: : :"

Authors' response: We changed it (Page 4 Line 22).

Page 4, line 21 and throughout the manuscript: Delete "value" in "value difference".

Authors' response: We modified it (Page 5 Line 8).

Page 5, line 24: Use a unique consistent term for particle number concentrations between 2.5 nm and 10 nm (either CN2.5-10 or CN2.5 – CN10) throughout the text!

Authors' response: Thanks! We used $CN_{2.5-10}$ in the revised manuscript (Page 5 Line 9; Page 5 Line 11; Page 5 Line 12; Page 5 Line 13; Page 5 Line 15).

Page 6, line 15: "speed" should be "loss rate".

Authors' response: We changed it (Page 6 Line 25).

Page 8, line 15: "should speculate" should be "indicates".

Authors' response: We changed it (Page 9 Line 1).

Page 9, line 18: Delete "(which is undefined days)".

Authors' response: We deleted it.

Page 9, line 23: Delete "whereas".

Authors' response: We deleted it.

Page 11, line 18-21: I do not understand the meaning of this phrase - please clarify: What is meant with "undefined case" here? Delete this line in Table 5.

Authors' response: We agree reviewer's opinion. We removed "undefined case" in Table 5 and text in the manuscript.

Page 12, line 2: ": : :a indicate decline: : :" should be ": : :indicate a decline: : :"

Authors' response: We changed it (Page 12 Line 16).

Page 12, line 3: ": : :discussed: : :" should be ": : :discussing: : :"

Authors' response: We changed it (Page 12 Line 18).

Page 12, line 4: ": : :simulation: : :" should be ": : :model: : :"

Authors' response: We changed it (Page 12 Line 19).

Page 12, line 7: Misleading phrase. The term "estimates of the biological characteristics" is somewhat vague, please specify.

Authors' response: We have replaced "estimates of the biological characteristics" to "estimates of the biological activities" (Page 12 Line 21)

Page 12, lines 8-9: DMS oxidation to sulphuric acid occurs in the atmosphere but not in the ocean – please correct!

Authors' response: We have replaced "oxidation of DMS in oceans" to "oxidation of DMS emitted from oceans" (Page 12 Line 23)

[revised manuscript text omitted]

---

## Author Comment (AC2) · 10 Apr 2019

We thank Referee 2 for providing valuable suggestions that improved the readability of our revised manuscript. Our responses to this Referee's major and minor points are stated below. The revised manuscript was uploaded in the form of a supplement.

Review comment on "New particle formation events observed at King Sejong Station, Antarctic Peninsula – Part 1: Physical characteristics and contribution to cloud condensation nuclei" by Jaeseok Kim et al. This manuscript presents new particle formation (NPF) and its impact on CCN ability at Korean Antarctic research Station (King Sejong) located in the Antarctic Peninsula. This study is based on long-term aerosol measurements for several years. To our knowledge, the long-term SMPS measurements through the years in the Antarctic regions are very limited. Actually, results in the manuscript are important and interesting to understand NPF and aerosol science in the Antarctic regions. As a whole, the topic of the manuscript is relevant and suitable for the scope of the "Atmos. Chem. Phys.". However, there are several points which require some careful revision and corrections before publication.

Major points
1. Authors showed NPF occurrence and frequency in Section of 3.1.1. However, time series of CN concentrations and SMPS results (i.e. contour plots of variations of aerosol size distributions) should be shown and add explanation before analysis/ discussion of NPF occurrence and frequency. The plots of the typical examples can provide important information for us.

Authors' response: Authors agree with the referee's comment. According to previous studies, time series of aerosol size distributions were showed for reader's understanding. Thus, examples of contour plots of aerosol size distributions is added in the revised manuscript (Figure 1).

[Figure]

Figure 1. Example of types of the NPF based on the SMPS data. (a) type A (18 January 2011-20 January 2011), (b) type B (13 January 2015) and (c) type C (9 January 2015). Type A is days when the formation and growth of nanoparticles should be clear. Type B is days when the formation occurred but growth was not clear. Type C is days when it cannot be said whether there is an event or not.

2. It is true that emission of aerosol precursors from oceanic bioactivity and atmospheric photochemical reactions are associated with NPF in the Antarctic coasts during summer. Unlike to other Antarctic coastal regions, however, anthropogenic impacts (local contamination) can be larger around the Antarctic Peninsula particularly in the summer because of activity in many stations and ship-borne tourism. Therefore, influence of anthropogenic activity and local contamination should be analyzed and discussed before discussion on contribution of condensable vapors originated from oceanic bioactivity. The following works are useful references.

Shirsat, S. V. and Graf, H. F.: An emission inventory of sulfur from anthropogenic sources in Antarctica, Atmospheric Chemistry and Physics, 9(10), 3397–3408, 2009.

Graf, H.-F., Shirsat, S. V., Oppenheimer, C., Jarvis, M. J., Podzun, R., and Jacob, D.: Continental scale Antarctic deposition of sulphur and black carbon from anthropogenic and volcanic sources, Atmospheric Chemistry and Physics, doi:10.5194/acp-10-2457-2010, 2010.

Authors' response: Authors agree with the referee's comment. Anthropogenic activity and local contamination do affect the characteristics of Antarctic ambient aerosols, including the NPF events. To minimize the effect of local contamination during the data analysis, we used black carbon concentration, wind speed and wind direction data. We described the methods to minimize the effects of local contamination in section of 2.2. The observatory is located approximately 400m southwest of the main buildings (includes a power generator and crematory). Thus, the northeastern direction (355–55°) is designated as a local pollution sector due to emissions from the power generator and crematory. Data collected from this sector are discarded. In addition, black carbon concentrations were measured simultaneously using an Aethalometer. Details of the Aethalometer measurements were described in detail in the previous work (Kim et al. 2017). Briefly, when black carbon concentration is higher than 100 ng m$^{-3}$, data were also excluded from analysis.

Furthermore, air masses in the Antarctic Peninsula were transported frequently from south America. This transport pathway can lead to high aerosol number concentrations and BC concentrations at Ferraz Station located in the Antarctic Peninsula (Pereira et al., 2004, 2006). In other words, these studies implied that anthropogenic aerosol precursors and land-origin aerosol precursors such as organics can be transported and supplied to the Antarctic Peninsula. Thus, I recommend strongly comparison of number concentrations, NPF frequency, and FR in each air mass origin.

Pereira, K., Evanhelista, H., Pereira, E., Simões, J., Johnson, E., and Melo, L.: Transport of crustal microparticles from Chilean Patagonia to the Antarctic Peninsula by SEMâˇARˇ EDS analysis, Tellus B, 56(3), 262–275, doi:10.1111/j.1600-0889.2004.00105.x, 2004.
Pereira, E., Evangelista, H., Pereira, K., Cavalcanti, I., and Setzer, A.: Apportionmentof black carbon in the South Shetland Islands, Antarctic Peninsula, Journal of Geophysical Research: Atmospheres (1984–2012), 111(D3), doi:10.1029/2005JD006086, 2006.

Authors' response: In table 5, NPF day number and FR were compared according to origin and pathway of air masses. The frequency of the NPF events of air masses originating from South America (Case I) was too low (only 3 days in this study) compared with other cases. Out of 101 NPF cases, only 3 cases were categorized as the cases when air masses came from South America. Because it is not meaningful to represent frequency and FR of the NPF events of air masses out of only 3 cases (Case I of the table), their analysis results are not shown in the manuscript.

3. Authors stated definition and classification of NPF in Section of 2.2.1 and 2.2.2. Because SMPS measured size distributions of aerosol particles with size range of D>10 nm, authors tried likely to identify NPF using the difference of CN concentrations (e.g., CN2.5-CN10). Criteria values of 500 cm-3 were used for the NPF identification. What is the procedure to decide the criteria values? This criteria is very important basic in this study. I think that authors were in accordance of procedures shown by Humphrires et al. (2016). Considering that measuring site and conditions were different to sea ice area (Humphrires et al., 2016), authors should show example plots of time series of CN2.5-CN10 and discuss the suitable criteria values. In addition, classification of NPF in accordance with previous works (Dal Maso et al., 2005; Yli-Juuti et al., 2009) requires information about particle growth after NPF. However, the difference of CN concentrations cannot provide information on particle growth. How did you identify particle growth of nucleation mode (D<10 nm)?

Authors' response: In the previous study (Kim et al., 2017), authors compared seasonal variations of CN concentrations between 2009 and 2015. Average $CN_{2.5-10}$ concentration was approximately 430 cm$^{-3}$ over the whole periods. Based on these results (not shown in the text), we used value of $CN_{2.5-10}$ of 500 cm$^{-3}$ as an emphirical condition of the NPF events. This first filtering process has made the selection of NPF more conservative and reliable before we go for the next condition of the NPF occurrence. Next process was using, $CN_{2.5-10}/CN_{10}$ values as a key parameter. The $CN_{2.5-10}/CN_{10}$ values can be used to distinguish between newly formed particles and background particles events (Warren and Seinfeld, 1985; Covert et al., 1992; Humphries et al., 2015).

For the identification of growth of nucleation mode particles, we cannot detect particle growth of particles less than 10 nm because only CN data and size distribution from 10 nm were available in this work. In this study, we considered particles smaller than 10 nm in diameter as newly formed particles, and for the calculation of growth we used SMPS data size distribution data ranging from 10 to 25 nm in diameter.

4. FR was estimated from CN data with 1 sec resolution in this study. What is values of variability? In general, 1 sec CN data can be varied greatly. The large variability engender the large error of the estimated FR. CN data with longer time resolution (e.g., one minute) is better to estimate FR. Also, statistical analysis and error estimation are required for CN2.5-CN10 and the estimated FR.

Authors' response: Because CN data with 1 s time resolution are highly fluctuating, , the FR in this study was estimated using average CN data per one minute.

To clarify we modified sentence to following text on Page 6 Line 2:
*"On the basis of the average number concentration data with 1 min time resolution, the FR was calculated for cases in which $_{CN2.5-10/CN10}$ values and $_{CN2.5-10}$ concentrations sharply increased (Fig. S1 in the Supplement)"*

5. GR was estimated from GMD. How did you calculate GMD? Did you have log-normal fitting analysis or identify diameter of mode maximum? Some explanation is needed in Section of 2.2.3.

Authors' response: GMD was calculated using log-normal fitting analysis.

We added following text Page 6 Line 18:
*"Here, the GMD was calculated from log-normal fitting analysis."*

6. To identify origins and pathway of air masses with NPF, some trajectory was shown in Figure 1. Although trajectory can provide us important information of transport processes of air masses, Fig.1 showed only some cases. I suggest all trajectories in NPF at each height are plotted in Figure 1 (e.g., trajectory density map) to identify origins and pathway of air masses with NPF.

Authors' response: In Figure 2, we showed example of the four cases with a steady air mass origin for each heights lasting during the NPF event periods, to highlight the fact that NPF cases were selected when steady air masses with similar origin. The origin of air masses arriving at the observation site during the NPF events (a total of 101 event days) was manually categorized into four cases by analyzing 48-h backward trajectory data ending at height of 100, 500 and 1500 m above the ground level. To comply the referee's suggestion, because 2-days trajectories can't be classified in four cases based on our category method, 99-days backward trajectories in 101 NPF event days can be shown in Figure 2. This figure can be shown in the Supplement (Fig. S4).

We added this sentence Page 12 Line 7:
*"Each trajectory according to four cases can be shown in Fig. S4 in the Supplement."*

[Figure]

Figure 2. 48-h air mass backward trajectories at height of (a) 100m, (b) 500 m and (c) 1500 m above the ground level of the sampling site. Because 2-day trajectories can't be classified in four cases based on category method in this study, 99-day trajectories were shown. Red, blue, pink and cyan colored line indicate that air masses originated from the South America area (Case I), Weddell Sea (Case II), Antarctic Peninsula area (Case III) and Bellingshausen Sea (Case IV), respectively.

7. CCN concentrations were discussed in Section of 3.3. Long-term CCN records provide important knowledge to us. In this study, aerosol size distributions were measured simultaneously by SMPS. Nevertheless, aerosol size distributions did not compare to CCN data. I understand that critical diameter was estimated hardly in this study. However, aerosol size distributions must be useful and important data to elucidate features of CCN concentrations. The critical diameter of the Antarctic aerosols during summer was discussed by Kyrö et al. (2013). Comparison between size distributions and CCN should be shown and discussed.

Authors' response: In previous studies (Pierce et al., 2014; Shen et al., 2016; Rose et al., 2017), the relationship between the NPF event and CCN concentration was determined by comparing number concentrations of particles larger than 50, 80 and 100 nm estimated by SMPS data are compared with aerosol size distribution data. In this study, whereas, CCN concentration measured directly by CCN counter were compared concentration of newly formed particles ($CN_{2.5-10}$) as the function of time during NPF event periods. Since it was very rare when the all 3 instruments – CPCs, SMPS, and CCN counter – are running together with the very best condition during the particle burst event, authors decide to choose the best way available, comparing CPC data with CCN during the 34 days with two dataset are available. In this manuscript, authors want to show the results that the CCN concentration increase are noticed for a couple of hours following NPF event under clean Antarctic environment, and this results are derived directly from in-situ CCN measurements.

Minor points
1. Introduction: Page 2 Line 20 Aerosol particles with size larger than several tens nm are not "new".

Authors' response: We removed "new" in text. (Page 2 Line 20).

2. Introduction: Page 3 Line 3 "Dall'osto" is correct.

Authors' response: We corrected it.

3. Introduction: Page 3 Line 10-12 Asmi et al. (2010) presented hygroscopicity of ultrafine particles measured at the coastal Antarctic station (Aboa). They showed and discussed hygroscopic growth factor and CCN activity, although they did not measure directly CCN. This should be mentioned in introduction.

Authors' response: We added following sentence in Page 3 Line 7:
*"Although CCN concentrations were indirectly estimated at Aboa, Asmi et al. (2010) also showed and discussed hygroscopic growth factor and CCN activity."*

4. Section 2.1 Measuring periods should be mentioned in Methods section, although the periods was shown in the section of Results and discussion.

Authors' response: We added periods in Section 2.1.

5. Page 8 Line 14-18 Kyrö et al. (2013) showed emission of aerosol precursors from melt pond, not from oceanic bio-activity. This description should be modified. "_biota activities in the Antarctica" is correct.

Authors' response: We have replaced *"..... along with precursor vapors derived from marine biota activities in the Antarctica (Virkkula et al., 2009; Kyrö et al., 2013; Weller et al., 2015; Jang et al., 2018)."* to *".... along with precursor vapors derived from marine biota activities in the Antarctica (Virkkula et al., 2009; Weller et al., 2015; Jang et al., 2018)."* (Page 9 Line 2)

6. Page 8 line 24-25 In this study, the NPF was observed in May in spite of only one case. If NPF occurred actually in the Antarctica in May, this is important to understand aerosol science in the Antarctic troposphere. Some explanation and discussion such as FR and air mass origin should be added.

Authors' response: Monthly variations in the FR during the NPF event periods were compared in Figure 4(a). Because the NPF event was observed one case in May, explanation and discussion of result were omitted in this analysis. However, according to referee's suggestion, in the revised manuscript, we modified sentence on Page 9 Line 10:

*"Although the FR was 0.20 cm$^{-3}$ s$^{-1}$ and air masses were probably originated from South America (Case I) in May, only one NPF event occurred."*

7. Figures 3 and 4 Both figures can be merged. That is easy to compare among each other.

Authors' response: To compare monthly characteristic of the NPF events, it is right to merge both figures (3 and 4). However, the way to estimate formation rate (FR) was different compared with estimation of growth rate (GR), condensation sink (CS) and source rate of condensable vapor (Q). The FR were calculated using CPC data, whereas the GR, CS and Q were estimated using SMPS data. To reduce confusion, authors used two figures. In the revised manuscript, however, we merge the two figures into one figure according to referee's opinion to easy compare among each other. In revised manuscript, we showed this figure as Figure 4.

[Figure]

Figure 3. Monthly variations of (a) the formation rates (FR), (b) the growth rates (GR) of nucleation mode particles ranging from 10 nm to 25 nm, (c) the condensation sink (CS), and (d) the source rate of condensable vapor (Q). The error bars represent a standard deviation.

8. Page 9 Line 22-24 GRs in September-October were not shown in Fig. 4. Does it mean no particle growth in September-October? Some explanation should be added.

Authors' response: GR values were calculated using SMPS data as mentioned section of 2.2.3. Unfortunately, SMPS data were unreliable owing to trouble of an instruments in September and October during the NPF event periods. Thus, the GRs in September and October were not shown in the manuscript.

We added following sentence in Page 10 Line 11:
*"The GRs in September and October were not shown due to mechanical trouble of the instruments."*

9. Page 10 Line 9-10 Higher CS values were obtained at King Sejong Station. The high CS might result from high aerosol number concentrations, although high CN related also to aerosol size distributions. If so (high aerosol concentrations), supply and transport of aerosols and aerosol precursors should be taken into account. This must be associated with FR, GR, and CCN ability. Details were already shown in the major comment.

Authors' response: Authors agree with referee's opinion. Anthropogenic and local impact can have an effect on high aerosol number concentrations. In this study, we also measured black carbon concentrations using Aethalometer. Based on the black carbon data, results including anthropogenic and local impact were discarded during analysis. When the black carbon concentrations were higher than 100 ng m$^{-3}$, aerosol number concentration and CCN data were excluded from analysis. In addition, data for wind speed and direction were used to minimize anthropogenic and local impact. The northeastern direction (355–55°) is designated as a local pollution sector due to emissions from the power generator and crematory. Data collected from this wind direction are discarded. Besides, when wind speed was less than 2 m s$^{-1}$, all data were also removed.

10. Page 11 Line 19-22 Are air mass origins (Case I-IV) corresponding to Fig.1a-d?

Authors' response: Yes, it is.

11. Section of 3.4 CS values were used for discussion. I suggest that CS values and aerosol number concentrations obtained in previous works at stations (e.g., Neumayer and Aboa) around Weddle Sea should be compared to data in this study. As mentioned in major comment, anthropogenic and local impact should be discussed. Such impacts are analyzed hardly only by trajectory.

Authors' response: For referee's suggestion, we tried to compare aerosol concentrations with other stations (e.g., Neumayer and Aboa) around Weddle Sea. However, it was difficult to compare the aerosol number concentrations due to limitation of data shown in papers. Weller et al. (2015) estimated CS values using light scattering data measured at Neumayer station and showed aerosol number concentrations during whole observation periods. In addition, Kyrö et al. (2013) introduced only median CS values during the entire campaign. As mentioned earlier in minor point 9, To minimize anthropogenic and local impact, in the present work, we used black carbon concentration, wind speed and wind direction data

[revised manuscript text omitted]

---

## Author Response (AR2)

Review comment on "New particle formation events observed at King Sejong Station, Antarctic Peninsula – Part 1: Physical characteristics and contribution to cloud condensation nuclei" by Jaeseok Kim et al.

The authors attempted valiantly to answer the points by the reviewers. Although the authors have succeeded in answering some points, our common and major criticism still remains. I (referee #2) did not call for more field measurements but for a better and clearer discussion. In the current revision, this is better and clearer, but some points of the revised manuscript are less clear.

Major comment
1. Impact of human activity in the Antarctic Peninsula in the summer
Generally, local contamination can affect strongly aerosol properties, particular in the Antarctic area as a pristine region. Therefore, we must remove locally contaminated data before analysis and discussion. Actually, authors attempted to do it in accordance with the procedure for data screening stated in the revised manuscript. However, this procedure can filter only the direct-contamination from the station (I mean King Sejong Station, here). Because local anthropogenic impact on BC concentrations depends on distance from the combustion source (Hagler et al., 2008), threshold value of 100 ng m-3 might be high for the contamination from other stations located in the Antarctic Peninsula.

Considering many ship-borne tourism and, any operating stations in the Antarctic Peninsula during summer, these impacts should be considered and discussed. Indeed, several stations are operating around King Sejong Station (King George Island). I understand that the authors removed hardly these impact from the measured data in this study. As shown by Shirsat et al (2009) and Graf et al (2010), however, human activity in the Antarctic Peninsula can have potential for atmospheric sulfur chemistry, particularly during summer. Additionally, BC concentrations at Ferraz Station (near King Sejong Station) were higher than those at other coastal stations (Pereira et al., 2006; Weller et al., 2013). These studies implies impact of human activities in the Antarctic Peninsula, although we must consider contribution of long-range transport from South America and latitudinal BC distribution. Thus, this is likely the current condition around the Antarctic Peninsula (in summer), although authors want to know aerosol properties in pristine conditions.

In the revised manuscript, higher aerosol concentrations, FR, and CS were observed at King Sejong Station. We can consider the following likelihoods for them; (1) distance from open sea surface, (2) oceanic bioactivity, (3) influence by human activity in the Antarctic Peninsula, and (4) long-range transport from South America. Therefore, some explanation and discussion about impact by human activity in the Antarctic Peninsula should be added into the text.

Hagler, G., Bergin, M., Smith, E., Town, M., and Dibb, J.: Local anthropogenic impact on particulate elemental carbon concentrations at Summit, Greenland, Atmospheric Chemistry and Physics, doi:10.5194/acp-8-2485-2008, 2008.
Pereira, E., Evangelista, H., Pereira, K., Cavalcanti, I., and Setzer, A.: Apportionment of black carbon in the South Shetland Islands, Antarctic Peninsula, Journal of Geophysical Research: Atmospheres (1984–2012), 111(D3), doi:10.1029/2005JD006086, 2006.
Weller, R., Minikin, A., Petzold, A., Wagenbach, D., and König-Langlo, G.: Characterization of long-term and seasonal variations of black carbon (BC) concentrations at Neumayer, Antarctica, Atmospheric Chemistry and Physics, doi:10.5194/acp-13-1579-2013, 2013.

Authors' response: We agree with referee's opinion. Although we applied strict rule to minimize effect of local contamination as mentioned in the manuscript, local pollution by human activities cannot be ruled out to be a potential factor to contribute the higher aerosol FR and concentrations. Because six stations are located within

a 10 km radius of sampling site, anthropogenic factors may influence concentrations and formation of atmospheric aerosols. We added text about effect of human activity in the revised manuscript on Page 10 Line 3.

*"Besides, human activities should be one of the possible reasons of high aerosol FR and concentrations. Although strict data filtering procedure was applied to the raw data-set to minimize the effect of local contamination as mentioned in Section 2.2, previous study showed that BC concentrations at King Sejong Station were higher than those at other stations in Antarctica (Kim et al., 2017). In fact, other studies (Shirsat and Graf, 2009; Graf et al., 2010) also reported that there were local pollution sources from tourist ships and emissions associated with scientific actives in Antarctic Peninsula, especially during austral summer seasons. These periodic human activities around the Antarctic Peninsula cannot be ruled out to be a potential factor to contribute the higher aerosol FR and concentrations."*

2. Comparison between size distribution and CCN

NPF is important aerosol source even in the Antarctic atmosphere. After growth to size of critical diameter, aerosol particles can act as CCN. If CCN concentrations depended on time after NPF as shown in Figure 5, this might relate to growth of nucleated particles. In other words, the normalized CCN variation can be varied in NPF types (Type A - C). In particular, authors can compare among aerosol number concentrations larger than critical diameter (ca. 50 nm), particles growth (change of size distribution after NPF), and CCN concentrations. Because authors had excellent data set of aerosol size distributions and CCN concentrations, comparison between size distribution and CCN can provide useful and valuable knowledge to us.

Authors' response: The aim of the section 3.3 'CCN concentration during NPF events' is to show the increasing pattern of measured CCN concentration (at 0.4% supersaturation) when NPF is observed. The authors are analyzing direct comparison between aerosol size distribution (SMPS data) and CCN concentration (CCNC data) for various meteorological conditions and air mass origins, not only the NPF cases. This further analysis using the long-term SMPS and CCNC data set will make a follow-up work. To comply with the referee's comment and respect the scope of the section 3.3, authors calculated particle concentrations larger than diameter 50, 80, 100 nm only for NPF cases as a function of time (hour) elapsed after the event. For this further analysis, authors had to limit the number of cases when all the three data-set (CPC, SMPS, CCNC) are available, which resulted the number of cases is 27. This result (with error bar) is shown as Figure 5, and text in Section 3.3 has been modified accordingly.

*"In this section, the contribution of particle formation to the variation in CCN concentration is investigated. Although recent studies reported that number concentrations of climate-relevant particles increased during NPF events (Pierce et al., 2014; Shen et al., 2016; Rose et al., 2017), the contribution of NPF to CCN concentration was estimated by using an indirect method. The number concentrations of particles larger than 50, 80 and 100 nm were estimated by using size distribution data. That value was considered as potential CCN concentration at different supersaturation value. In the present study, CCN concentrations at a supersaturation value of 0.4% were directly measured using CCN counter. Hourly mean CCN concentrations were compared with CN concentrations measured by CPC and size distribution results measured by SMPS (Fig S3 in the Supplement). Data for only 27 days, when all the three data-set (CPC, CCN counter, and SMPS) were available, were analyzed. Fig. 5 shows variation in $CN_{2.5-10}$ concentrations, CCN concentrations, and number concentrations as a function of time elapsed after the NPF event. The zero in the x-axis means the start time of the NPF event. As shown in Fig. 5a and b, the $CN_{2.5-10}$ concentrations sharply increased at NPF start time and the peak concentration occurred 2 h afterward, whereas the CCN concentrations gradually increased for 8 h.*

*Indeed, the maximum CCN concentrations rose from 191.4±16.3 cm⁻³ to 213.2±17.7 cm⁻³ before and after the NPF events, respectively, showing an increase of 11%. Fig 5b also shows variation of number concentrations ($N_{50}$, $N_{80}$, and $N_{100}$) of particles larger than 50 nm, 80 nm, and 100 nm, respectively. Number concentrations were calculated from aerosol size distribution data. Variation trends of the number concentrations were similar to those of CCN concentrations, increasing approximately 15% before and after the NPF events."*

[Figure]

Figure 5. Variation in (a) $CN_{2.5-10}$ concentrations measured using CPC and (b) CCN concentrations measured with CCN counter and number concentrations calculated using SMPS data with time. $N_{50}$, $N_{80}$, and $N_{100}$ represent number concentrations of particles lager than 50 nm, 80 nm, and 100 nm in diameter, respectively. The zero in the x-axis indicates the start time of the NPF events.

Minor comments

1. Page 10 line 20: the ultrafine particles of <100 nm in diameter can...

Authors' response: Authors changed text.

2. Procedures of log-normal fitting (I mean that equation) should be mentioned and/or earlier works should be referred.

Authors' response: Authors added reference for procedures of log-normal fitting in the revised manuscript on Page 6 Line 19.

*"Here, the GMD was calculated from log-normal fitting analysis (Hinds, 1999)."*

3. Specific values of diffusion coefficient (H2SO4) and transitional regime correction factor should be added in the text. These descriptions are helpful for readers.

Authors' response: Authors added specific values of diffusion coefficient (0.1 $cm^2 s^{-1}$) in the revised manuscript on Page 7 Line 8, whereas transitional regime correction factor is not added because it is related to particle size.

*"where D is the diffusion coefficient of the condensable vapor (0.1 cm2 s-1), β is the transitional regime correction factor from Fuchs and Sutugin (1970),"*

4. Fig. 4: No data of GR, CS, and Q were mentioned in the text. I recommend that no data are marked by symbols such as asterisks and short explanations are added in the figure caption.

Authors' response: Authors added short explanations in the Figure 4 caption.
*"No NPF events were observed in June, July, and August. The GRs, CSs, and Q values in September and October were not shown due to mechanical troubles of the instruments."*

5. Figure 5: Error bars should be shown in both plots.

Authors' response: Authors added error bars in Figure 5.

6. Table 3: I recommend that the parameters, FR, GR, CS, and Q, in Case I are shown in Tab. 3. Because of low data number, min-max of the parameters are useful for us (readers).

Authors' response: In revised manuscript, for Case I, authors added parameters such as the FR, the GR, the CS and the Q in Table 3.